# Protective Role of Indole-3-Acetic Acid Against *Salmonella* Typhimurium: Inflammation Moderation and Intestinal Microbiota Restoration

**DOI:** 10.3390/microorganisms12112342

**Published:** 2024-11-16

**Authors:** Yuxin Fan, Qinglong Song, Siyu Li, Jiayu Tu, Fengjuan Yang, Xiangfang Zeng, Haitao Yu, Shiyan Qiao, Gang Wang

**Affiliations:** 1State Key Laboratory of Animal Nutrition and Feeding, College of Animal Science and Technology, China Agricultural University, Beijing 100193, China; fyuxinfan@163.com (Y.F.); li062528@163.com (S.L.); tujiayu0302@163.com (J.T.); yangfengjuan@cau.edu.cn (F.Y.); ziyangzxf@163.com (X.Z.); yuhaitao@cau.edu.cn (H.Y.); qiaoshiyan@cau.edu.cn (S.Q.); 2Beijing Key Laboratory of Biological Feed Additive, China Agricultural University, Beijing 100193, China; 3Frontier Technology Research Institute of China Agricultural University in Shenzhen, Shenzhen 518116, China; sql19972002@126.com

**Keywords:** indole-3-acetic acid, *Salmonella* typhimurium, intestinal barrier, colonic microbiota

## Abstract

Indole-3-acetic acid (IAA), a metabolite derived from microbial tryptophan metabolism, plays a crucial role in regulating intestinal homeostasis. However, the influence and potential applications of IAA in the context of animal pathogen infections remain underexplored. This study investigates the prophylactic effects of IAA pretreatment against *Salmonella* typhimurium (ST) SL1344 infection, focusing on its ability to attenuate inflammatory responses, enhance intestinal barrier integrity, inhibit bacterial colonization, and restore colonic microbiota dysbiosis. The results demonstrated that IAA ameliorated the clinical symptoms in mice, as evidenced by reduced weight loss and histopathological damage. Furthermore, IAA inhibited the inflammatory response by downregulating the gene expression of pro-inflammatory cytokines *IL-17A*, *TNF-α*, *IL-1β*, and *IL-6* in colon, ileum, and liver. IAA also preserved the integrity of the intestinal mucosal barrier and promoted the expression of tight junction proteins. Additionally, 16S rRNA gene sequencing revealed significant alterations in intestinal microbiota structure induced by ST infection following IAA treatment. Notable changes in β diversity and species richness were characterized by the enrichment of beneficial bacteria including Bacteroideaceae, Spirillaceae, and Bacillus. The proliferation of *Salmonella enterica* subspecies *enterica* serovar Typhi was significantly inhibited, thereby enhancing the intestinal health of the host. In summary, the oral administration of IAA contributes to the alleviation of inflammation, restoration of the intestinal barrier, and correction of colonic microbiota disturbance in mice challenged with ST.

## 1. Introduction

*Salmonella* is a prevalent foodborne pathogen capable of infecting and causing disease in humans, livestock, mice, birds, and wild animals [1]. Globally, *Salmonella* infections frequently rank as the leading cause of bacterial food poisoning [2]. *Salmonella* is estimated to cause more than 300,000 deaths annually [3]. In China, reports indicate that 70% to 80% of bacterial food poisoning cases are attributed to *Salmonella* [4].

*Salmonella* typhimurium (ST) represents a highly adaptable pathogen that constitutes a considerable risk to public health, serving as a principal etiological agent of gastroenteritis in humans and systemic disease in susceptible murine models [5,6]. This pathogen has been associated with the disruption of intestinal epithelial barrier integrity, alterations in the intestinal microbiota, induction of inflammatory responses, and interactions with dietary and environmental factors [7]. It is essential to investigate both endogenous and exogenous bioactive compounds that enhance systemic anti-inflammatory responses [2] and promote intestinal homeostasis [8,9] to improve recovery from *Salmonella* infection.

Recent studies have elucidated that tryptophan catabolites generated by gut microbiota are crucial for maintaining gut homeostasis and modulating a range of host physiological functions [10,11,12]. The gut microbiota is integral to the management of pathogenic bacterial infections through the production of short-chain fatty acids, indole derivatives, and other bacterial metabolites [13,14]. One significant metabolite derived from dietary tryptophan is indole-3-acetic acid (IAA). Various bacteria such as *Bacteroides_fragilis*, *Clostridium_sticklandii*, *Bifidobacterium_adolescentis*, and *Eubacterium_cylindroides* in the host gut can metabolize dietary tryptophan, converting it into indole-3-aldehyde, indole-3-acetic acid, serotonin, indole-3-acetamide, indole-3-acetonitrile, and then converting it to IAA or directly metabolizing ortho aminobenzoic acid to produce indole-3-acetonitrile, which is then converted to IAA [11]. As an activator of aromatic hydrocarbon receptor (AHR) in intestinal immune cells and intestinal epithelial cells, IAA modulates both innate and adaptive immune responses in a ligand specific manner, and in vitro studies have shown its notable effects on the resistance of inflammation and oxidative stress, mucosal repair, and maintenance of homeostasis [15,16,17,18,19]. However, no available study has explored the protective effects of IAA against pathogenic bacterial infection in vivo.

Consequently, this study utilized a mouse model of ST SL1344 infection to examine the effects of IAA treatment on systemic inflammation, intestinal barrier function, and intestinal microbiota structure.

## 2. Materials and Methods

### 2.1. Ethics Statement

The experiment was approved by the Institutional Animal Care and Use Committee of China Agricultural University (Beijing, AW32013202-1-1). All protocols were performed according to the relevant standards of animal welfare of China Agricultural University.

### 2.2. Bacteria Preparation

ST SL1344 (with natural resistance to streptomycin) was grown in Luria–Bertani (LB) medium that contained Tryptone (10 g/L), Yeast extract (5 g/L), and NaCl (10 g/L) at 37 °C with shaking at 56× *g* for 12 h. The pellet was washed with phosphate-buffered saline (PBS) and adjusted to a density of 10^7^ colony forming units (CFU)/mL.

### 2.3. Materials and Preparation

Indole-3-acetic acid was purchased from Sigma-Aldrich (St. Louis, MO, USA). Mouse serum ELISA kits were purchased from CUSABIO Biotech Co., Ltd. (Wuhan, China).

### 2.4. Mouse Model and Treatments

Female BALB/c mice (*n* = 50; aged 8 to 10 weeks; SPF Biotechnology Co., Ltd., Beijing, China) were randomly assigned to five treatments (ten mice per treatment): control (CON), ST-infected (ST_CON), and three ST-IAA treatments (ST_IAA_40, ST_IAA_80, ST_IAA_160). During the entire trial period, control and ST-infected mice were gavaged with 0.3 mL of sterile physiological saline daily, while ST-IAA mice received daily gavaged of 0.3 mL of IAA aqueous solution at concentrations of 40, 80, and 160 mg/kg, respectively. After 7 days of IAA treatment, the ST-infected and ST-IAA mice were gavaged with 0.1 mL streptomycin (20% *w*/*v*), whereas the control mice were treated with 0.1 mL sterile saline. Following a 24-h period post-streptomycin administration, the ST-infected and ST-IAA mice were then gavaged with 0.2 mL ST (6.0 × 10^7^ CFU/mL).

### 2.5. Sample Collection and Serum Chemical Analysis

After 96 h of ST infection, the mice were euthanized by cervical dislocation. The serum, liver, ileum, colon, and colonic contents were collected for subsequent experiments. Serum levels of IL-1β, TNF-α, and IL-6 were measured according to the instructions provided in the indirect enzyme-linked immunosorbent assay (ELISA) kits supplied by Huamei Biotech (Wuhan, Hubei, China).

### 2.6. Intestinal Histopathological Evaluation

Fresh ileum and colonic tissues (1 cm) were collected and fixed in 4% paraformaldehyde, dehydrated gradually with ethanol from 85% (*v*/*v*) to 100% (*v*/*v*), and were stained with hematoxylin and eosin (HE). Fixed, stained tissues were observed with a light microscope (Olympus XC41, Tokyo, Japan) for histopathological changes. The pathological scores were assigned based on five criteria: inflammatory cell infiltration of mucosa and submucosa, lamina propria edema, acinar dilation, changes in the number of exfoliated goblet cells, and fibrosis. The specific scoring criteria are shown in Table 1. The comprehensive pathological score for each sample was determined by summing the scores across all criteria.

### 2.7. Quantitative Real-Time Polymerase Chain Reaction (PCR) Analysis

Total RNA was extracted from mouse colonic, ileal, and liver tissues using TRIzol reagent (Invitrogen, Carlsbad, CA, USA). Isolation of total RNA was accomplished using a previously described method [20]. Concentration of RNA was evaluated using a spectrophotometer (NanoDrop 2000, Thermo Scientific Co., Ltd., Wilmington, DE, USA). One μg of RNA was used to generate cDNA at volumes of 200 μL. Primers for real-time PCR are shown in Table 2. Glyceraldehyde-3-phosphate dehydrogenase (GAPDH) was used as an internal reference in this study. Relative mRNA expression of the target genes was determined using the 2^−∆∆Ct^ method [21,22].

### 2.8. Fecal Microbial Quantity and16S rRNA Bioinformatics Analysis

The microbial composition of colon contents in mice were analyzed using Illumina’s Miseq PE3000 high-throughput sequencing platform (Illumina, Santiago, CA, USA). The 16S rRNA gene in the V3–V4 region of bacteria was amplified using a thermal cycling polymerase chain reaction system (GeneAmp 9700, ABI, Foster City, CA, USA). We utilized QIIME 1.9.1 (http://qiime.org/scripts/assign_taxonomy.html, accessed on 14 October 2024) to demultiplex the original Illumina fastq files and filter their quality. Operational taxonomic units (OTUs) were defined as units with a similarity threshold of 97% using UPARSE. UCHIME was applied to identify and remove abnormal gene sequences [23]. Classification dependency analysis of OTUs was performed using the ribosome database project classifier at a 90% confidence level. The Sobs, Shannon, and Simpson diversity indices were calculated to reflect α diversity at 97% identity with Mothur software (version: 1.35.03) [24].

### 2.9. Statistical Analysis

All data were assessed for normal distribution and homogeneity of variance. Analyses were performed using SPSS using one-way analysis of variance. When significant differences between treatments were detected, Duncan’s method was utilized for multiple comparisons. GraphPad Prism 7.0 (GraphPad Software Inc., San Diego, CA, USA) was employed for graphical representation. Results are expressed at means ± standard error of the mean (SEM). The significance level was set as *p* < 0.05 to indicate significant differences, *p* < 0.01 for extremely significant differences, and 0.05 < *p* < 0.10 for statistical trends.

## 3. Results

### 3.1. Growth Performance and Survival Rate

The pretreatment of mice with IAA for 7 days prior to ST infection did not result in any sign of toxicity compared to normal mice, as evaluated by BW and food intake. ST-infected mice began to lose weight 24 h after ST infection (Figure 1A). However, treatment with ST_IAA_40 for 4 days significantly ameliorated body weight loss compared to ST_CON (*p* < 0.05; Figure 1A). In comparison to CON, ST_CON significantly reduced the survival rate of mice to 80% (Figure 1B). Conversely, treatment with ST_IAA_40 and ST_IAA_80 improved the survival of mice to 100% 4 days following ST infection compared to ST_CON, while ST_IAA_160 treatment had no significant effect on the survival rate of mice (Figure 1B).

### 3.2. Intestinal Histopathology

ST infection resulted in severe inflammation of the colon, characterized by mucosal edema, infiltration of inflammatory cells into the mucosal layer, edema of the lamina propria, reduction of goblet cell, exfoliated of cells in the intestinal lumen, crypt injury, obvious fibrosis, and macrophage infiltration by HE staining (Figure 2E). In contrast, ST_IAA_40 and ST_IAA_80 significantly mitigated this histopathological injury in both colon (Figure 2A,B) and ileum (Figure 2C,D). Compared to ST_CON, IAA treatment improved the infiltration of inflammatory cell and reduced lamina propria edema (Figure 2B,D; *p* < 0.05). Additionally, the acinar dilation and the decrease in goblet cells were significantly alleviated in the ST_IAA_40 and ST_IAA_80 groups (Figure 2B; *p* < 0.05).

### 3.3. mRNA Expression Levels of Pro-Inflammatory Cytokines

Compared with the CON group, the ST_CON group exhibited increased mRNA expression of *TNF-α*, *IL-1β*, and *IL-6* in both the colon (Figure 3A–C) and ileum (Figure 3D–F) tissues. However, IAA treatment significantly reduced the elevated expression of these cytokines caused by ST (*p* < 0.05). In comparison to the ST_CON group, ST_IAA_40 and ST_IAA_80 groups showed decreased mRNA expression of pro-inflammatory factors *TNF-α* (Figure 3A) and *IL-6* (Figure 3C) in the colon (*p* < 0.05). Similarly, the increased mRNA levels of pro-inflammatory factors *TNF-α* (Figure 3D), *IL-1β* (Figure 3E), and *IL-6* (Figure 3F) mRNA in the ileum were also reduced by ST_IAA_40 and ST_IAA_80 (*p* < 0.05).

ST infection can induce severe systemic infection and inflammation in tissues. In this study, the mRNA expression of pro-inflammatory factors, such as *IL-1β* and *TNF-α*, in the liver was found to be dramatically increased after infection. IAA treatment significantly attenuated the inflammatory response in the liver of mice. Mice assigned to ST_IAA_40 and ST_IAA_80 treatments expressed reduced mRNA levels of *TNF-α*, *IL-1β*, and *IL-6* compared to ST_CON mice (*p* < 0.05; Figure 3G–I).

### 3.4. mRNA Expression Levels of Tight Junction Protein

ST inhibited the mRNA expression of tight junction protein Occludin in intestine and liver compared to the CON group (Figure 4). IAA enhanced the mRNA expression of *Occludin* in both the colon (Figure 4A) and liver (Figure 4C; *p* < 0.05).

### 3.5. Th17 and ILC3 Transcription Levels of Intestinal Immune Cells

Moreover, the analysis of mRNA expression in adaptive immune cell analysis indicated that ST_IAA_40 regulated the transcription of Th17 cells and significantly downregulated the mRNA expression of *IL-17A* in the colon (Figure 5A) and Th17-specific transcription factor *RORγt* in the colon (Figure 5B; *p* < 0.05). In the ileum, compared to ST_CON, the transcription level of *IL-17A* in ST_IAA_40 and ST_IAA_80 was significantly down-regulated and restored to levels close to those of the CON group (Figure 5D). Additionally, the mRNA expression of *RORγt* in ST_IAA_40 was decreased (Figure 5E; *p* < 0.05). IAA had no significant effect on the mRNA expression of intestinal ILC3-specific transcription factor *NKp46* (Figure 5C,F).

### 3.6. Serum IL-1β, TNF-α and IL-6 Concentration

Serum levels of the pro-inflammatory factors indicated that IAA treatment reduced the systemic inflammatory response compared with ST_CON, and the ST_IAA_80 treatment significantly decreased serum IL-1β concentrations (Figure 6A; *p* < 0.05). Compared to the model group, IAA exhibited a decreasing trend in serum TNF-α and IL-6 levels (Figure 6B,C).

### 3.7. Colonic Microbiota Structure Changes

The structure of microbial communities influenced by ST infection in the colonic contents were analyzed. PCoA plots based on Bray–Curtis distances indicated significant differences in the composition of the colonic microbiota among the five treatments (ANOSIM; *R* = 0.359, *p* = 0.010; Figure 7A). Compared to ST_CON or CON, ST_IAA_40 markedly affected the composition of colonic microbiota (Figure 7A, *p* < 0.05).

The microbial biomarkers and the relative abundance of the colonic microbiota were analyzed. At the phylum level, Proteobacteria, Firmicutes, and Bacteroidota were identified as the dominant phyla. The relative abundances of Proteobacteria was strikingly increased in ST_CON group compared to others, while Bacteroidota levels in the CON group was higher than those in the ST groups (Figure 7B,C). Compared to the ST_CON treatment, ST_IAA_160 reduced the levels of Proteobacteria (*p* < 0.05) to levels close to those in CON (Figure 7B,C). At the genus level, ST_IAA_40 reduced *Salmonella* abundance compared to the ST_CON treatment (*p* < 0.05; Figure 7B,D). At the species level, different concentrations of IAA significantly reduced the abundance of *Salmonella enterica* subspecies *enterica* serovar Typhi compared to the ST_CON group (*p* < 0.05), with ST_IAA_40 exhibiting the most pronounced effect (Figure 7B). LEfSe analysis indicated that, compared to the ST_CON treatment, ST_IAA_40 up-regulated the abundance of *Cerasibacillus*, *Novosphingobium*, and *Ralstonia*, while *Salmonella* was identified as the key species in mice subjected to the ST_CON treatment (LDA > 2, *p* < 0.05; Figure 7E,F).

## 4. Discussion

Tryptophan-derived microbial metabolites such as IAA, indole-3-pyruvic acid, and indole-3-acetate have been shown to exhibit protective effects against intestinal inflammation [19,25,26,27]. However, there are scant in vivo studies on the antibacterial properties of IAA. In this study, a mouse model of ST SL1344 infection was developed to examine the potential impact of IAA administration on physiological changes in pathogen-infected mice. Our findings revealed that IAA administration mitigated the adverse effects associated with ST infection, notably ameliorating weight loss and reducing mortality rates in infected mice. Specifically, significant weight loss was observed in mice 24 h post-ST infection, followed by hematochezia, anorexia, chills, and even mortality, consistent with previous research findings [28,29,30]. As the experimental period progressed, symptoms severity escalated in both the ST-infected model group and the IAA-treated group. However, significant improvements were observed in physiological parameters among mice in the IAA-treated group. These findings highlight the potential therapeutic value of IAA in managing intestinal inflammation and gut pathogen infections.

The intestinal mucosal barrier functions as an essential defense mechanism, selectively permitting the absorption of nutrients and preventing the translocation of harmful substances into bloodstream, thereby preserving gastrointestinal health [31,32]. The structural integrity of the colonic epithelium was disrupted by ST infection, resulting in an ineffective intestinal barrier against microbial invasion [33]. This led to the activation of antigen-presenting cells, which subsequently triggered colonic inflammation, crypt damage, and a notable expansion of lamina propria edema and acinar dilation [7,34,35]. Conversely, IAA markedly mitigated structural damage to the colonic and ileal mucosa, upregulated the expression of tight junction proteins, and restored the intestinal epithelial barrier function. These results underscore the efficacy of IAA in ameliorating intestinal pathological alterations and enhancing intestinal barrier integrity.

ST infection elicits inflammation in the intestines [2,9]. Our findings demonstrate that ST infection dramatically upregulates the expression of proinflammatory cytokines TNF-α, IL-1β, and IL-6 in intestinal tissues, with TNF-α transcription in the liver reaching 120-fold compared to normal controls. Notably, IAA exhibited the most pronounced inhibitory effect on the elevation of TNF-α expression. TNF cytokines play a central role among inflammatory cytokines, acting as critical mediators of proinflammatory responses in intestinal inflammation [36,37]. IL-6 activates various target cells, including antigen-presenting cells and T cells, thereby intensifying the inflammatory response [38]. IL-1β acts as an activator of NF-κB, facilitating the secretion of proinflammatory cytokines by stimulating innate lymphoid cells, enhancing the recruitment of neutrophils, and amplifying the inflammatory response [39,40]. Our study demonstrated that IAA effectively reduced the transcription levels of these proinflammatory cytokines in both intestinal and liver tissues, as well as serum IL-1β levels. These findings substantiate the role and efficacy of IAA in alleviating intestinal inflammation and enhancing systemic immune function compromised by ST infection.

Intestinal homeostasis can be disrupted by medications, infections, and malnutrition [41]. The gut microbiota is linked to conditions like inflammatory bowel diseases, obesity, and irritable bowel syndrome, suggesting that microbiome manipulation could reduce disease risks and improve gut health [42,43]. Our findings reveal that oral administration of IAA effectively prevents and alleviates pathogenic bacterial infections, exerting a substantial impact on the diversity and composition of colonic microbiota. Specifically, IAA administration resulted in the downregulation of the abundance of *Salmonella enterica* subspecies *enterica* serovar Typhi, as well as other potentially pathogenic species, including *Bacteroides_cidifaciens*, Verrucomicrobiae, and Actinobacteriota. Behary et al. demonstrated that gut dysbiosis induces an immunosuppressive phenotype, with a significant enrichment of *Bacteroides_caecimuris* in non-alcoholic fatty liver disease-related hepatocellular carcinoma [44]. Additionally, the abundance of Verrucomicrobiae decreases in the gut microbiota of patients with irritable bowel syndrome undergoing dietary intervention, thereby contributing to the improvement of intestinal diseases [45]. The administration of IAA specifically enriched potential functional microorganisms, such as Bacillaceae [46], *Novosphingobium* [47], and Sphingomonadaceae [48]. These microorganisms possess the metabolic capability to convert tryptophan into indole derivatives, which may contribute to the maintenance of intestinal health [49].

In conclusion, IAA has the capacity to inhibit the colonization of intestinal pathogens, ameliorate pathological symptoms in mice infected with pathogenic bacteria, reduce intestinal and liver inflammation, decrease intestinal permeability, and restore barrier function. These findings substantiate that IAA exerts a broad inhibitory effect on intestinal pathogens, while also modulating systemic immunity and promoting intestinal health.

## Figures and Tables

**Figure 1 microorganisms-12-02342-f001:**
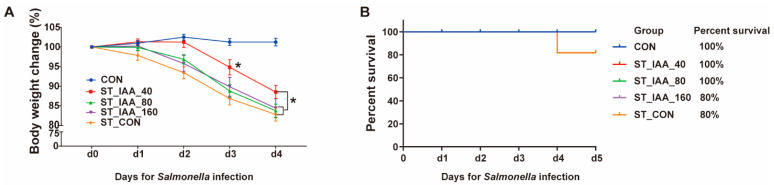
Effects of IAA supplementation on growth performance and survival rate of mice. Body weight change of mice after ST-infected 1, 2, 3, 4 days (**A**). Data were expressed as the mean ± SEM, *n* = 8, * *p* < 0.05. Compared with CON, the survival rate of mice in ST_IAA_40, ST_IAA_80, ST_IAA_160, and ST_CON after ST-infected 4 days (**B**), *n* = 10.

**Figure 2 microorganisms-12-02342-f002:**
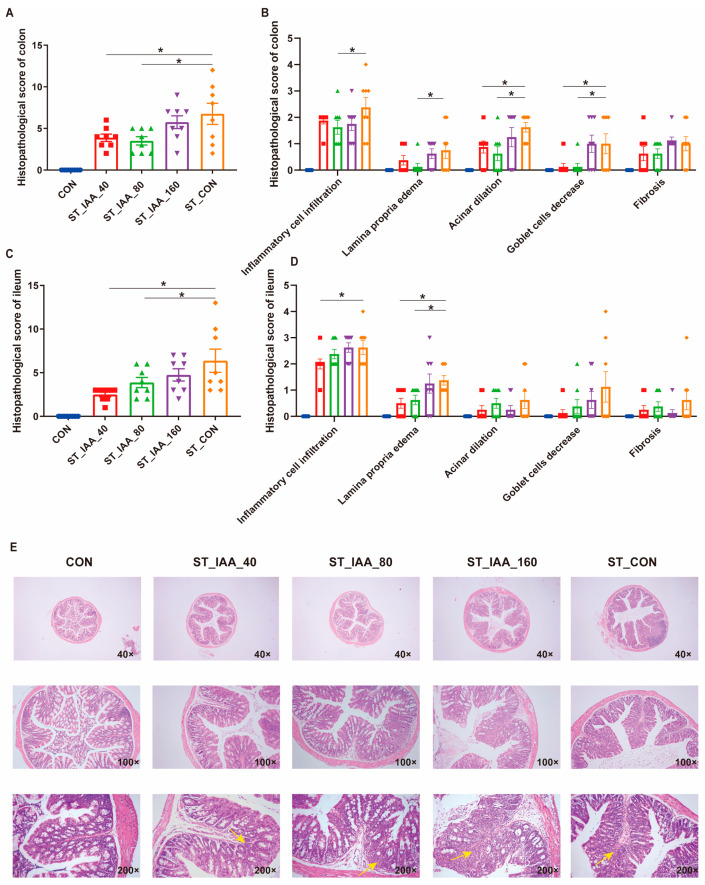
Effects of IAA supplementation on the histopathology of colon and ileum of mice. Summary of histopathological scores of colon (**A**) and ileum (**C**); inflammatory cell infiltration, lamina propria edema, acinar dilation, goblet cells decrease and fibrosis scores of colon (**B**) and ileum (**D**) tissue; histopathological changes of colon (×40, ×100, ×200; yellow arrows represent inflammatory cell infiltration caused by ST) (**E**). Data were expressed as the mean ± SEM, *n* = 8, * *p* < 0.05.

**Figure 3 microorganisms-12-02342-f003:**
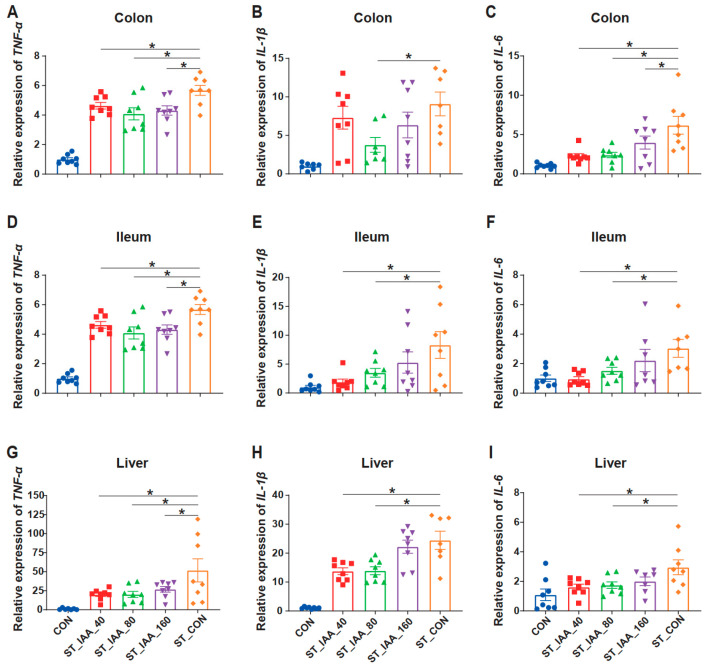
Effects of IAA supplementation on intestinal and tissue inflammation of mice. Relative mRNA expression levels of inflammatory cytokines *TNF-α*, *IL-1β*, and *IL-6* in colon (**A**–**C**), ileum (**D**–**F**) and liver (**G**–**I**). Data were expressed as the mean ± SEM, *n* = 7–8, * *p* < 0.05.

**Figure 4 microorganisms-12-02342-f004:**
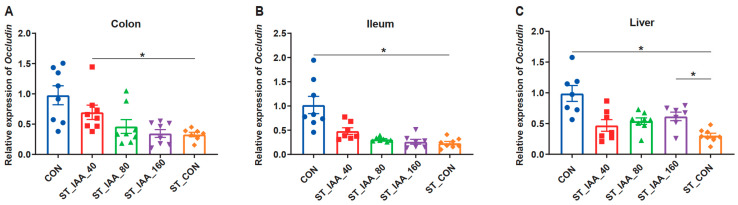
Effects of IAA supplementation on intestinal and liver tight junction protein expression of mice. mRNA expression levels of *Occludin* in colon (**A**), ileum (**B**), and liver (**C**). Data were expressed as the mean ± SEM, *n* = 7–8, * *p* < 0.05.

**Figure 5 microorganisms-12-02342-f005:**
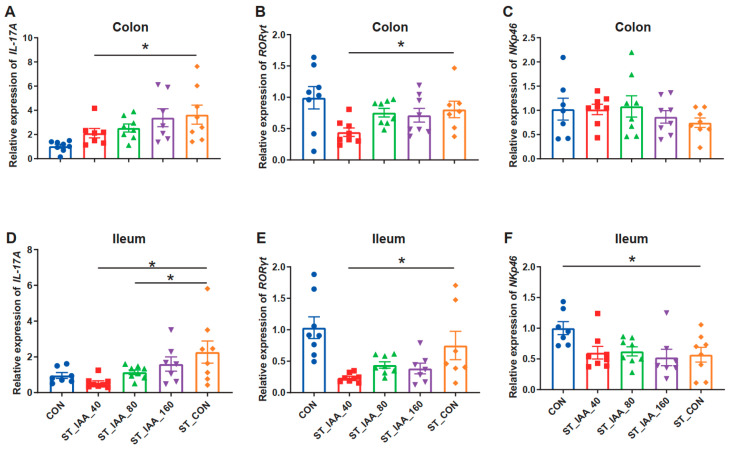
Effects of IAA on transcription of intestinal Th17 and ILC3 of mice. The mRNA expression of *IL-17A* (**A**,**D**), *RORγt* (**B**,**E**), and *NKp46* (**C**,**F**) in colon and ileum. Data were expressed as the mean ± SEM, *n* = 7–8, * *p* < 0.05.

**Figure 6 microorganisms-12-02342-f006:**
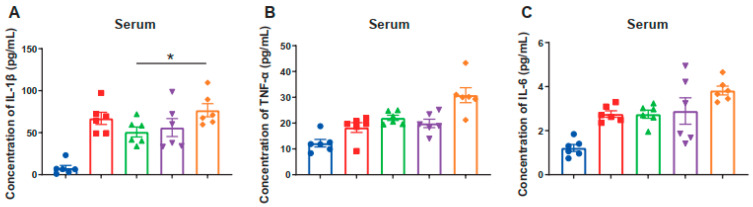
Effects of IAA on serum concentrations of proinflammatory cytokines in mice. Serum levels of inflammatory cytokine IL-1β (**A**), TNF-α (**B**), IL-6 (**C**) were measured by ELISA. Data were expressed as the mean ± SEM, *n* = 6, * *p* < 0.05.

**Figure 7 microorganisms-12-02342-f007:**
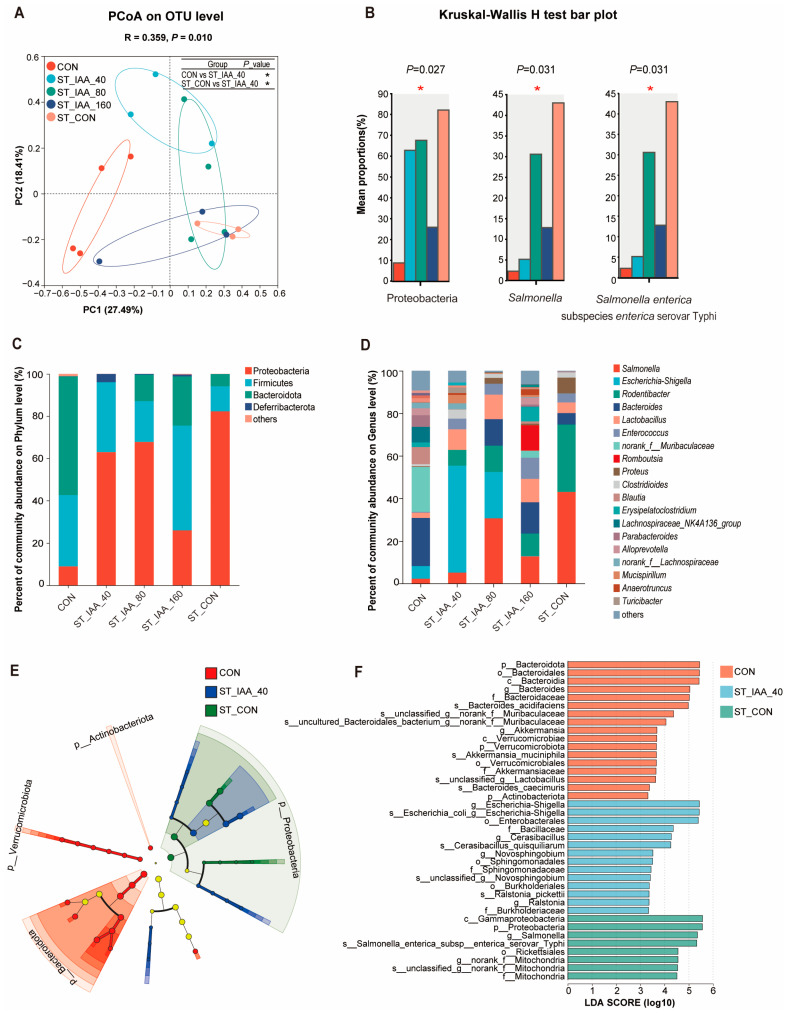
Effects of IAA supplementation on colonic microbiota of mice. PCoA analysis (**A**) of β diversity of colonic microbiota on day 4, * *p* < 0.05. Relative abundance of species at phylum, genus and species levels of colonic microbiota (**B**), * *p* < 0.05. Bar of horizontal community composition of phylum (**C**) and genus (**D**). Taxonomic cladogram from phylum to species and LDA score plot generated from LEfSe of 16S rRNA gene amplification sequencing data (LDA > 2, *p* < 0.05) (**E**,**F**).

**Table 1 microorganisms-12-02342-t001:** Criteria for intestinal histological damage score of ST infected mice.

Item	Inflammatory Cell Infiltration	Lamina Propria Edema	Acinar Dilation	Goblet Cells Decrease	Fibrosis
0	No lesions	No lesions	No lesions	No lesions	No lesions
1	Mild	Mild edema	Mild	Mild	Mild
2	Moderate, mucosa inflammatory cell infiltration	Marked edema	Marked expansion	Moderate, exfoliated cells	Moderate
3	Severe, inflammatory cells infiltration submucosa	Severe edema	/	Severe exfoliated cells	Severe
4	Severe, mucosal severe edema	/	/	/	/

Notes: / Indicates that no rating is performed.

**Table 2 microorganisms-12-02342-t002:** The primer sequences of target genes used for RT-qPCR.

Gene Name	Forward Primer Sequence (5′ to 3′)	Reverse Primer Sequence (5′ to 3′)	Size (bp)
*GAPDH*	AACTTTGGCATTGTGGAAGG	ACACATTGGGGGTAGGAACA	223
*TNF-α*	ACCCTCACACTCACAAACCA	GGCAGAGAGGAGGTTGACTT	246
*IL-1β*	TCAGGCAGGCAGTATCACTC	AGCTCATATGGGTCCGACAG	250
*IL-6*	CTGCAAGAGACTTCCATCCAG	AGTGGTATAGACAGGTCTGTTGG	131
*Occludin*	AAGTCAACACCTCTGGTGCC	TCATAGTGGTCAGGGTCCGT	173
*RORγt*	TCCTGCCACCTTGAGTATAGTC	GTAAGTTGGCCGTCAGTGCTA	80
*NKp46*	AATGGAAACTCGGTGAACATCTG	GGGGTTGCTCGACTTTGAC	216
*IL-17A*	ACTCTCCACCGCAATGAAGA	CTCTCAGGCTCCCTCTTCAG	161

## Data Availability

The original contributions presented in the study are included in the article, further inquiries can be directed to the corresponding author.

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
