# Peer review of "Protective Role of Indole-3-Acetic Acid Against Salmonella Typhimurium: Inflammation Moderation and Intestinal Microbiota Restoration"

_microorganisms, 2024, doi:10.3390/microorganisms12112342_

Round 1
Reviewer 1 Report
Comments and Suggestions for Authors
This is an interesting paper on the protective role of indole-3-acetic acid against Salmonella Typhimurium. However, I would like to draw your attention to the following:
Introduction: IAA presents biological functions in many organisms and has been studied mostly in plants since it acts as a plant growth hormone. Please refer in more detail on the production, possible bacterial catabolism and roles of IAA in the host gut and the common gut bacteria.
Results: Figure 1: more details are needed in legend. Can B, C, D, E be combined in one graph? Also, all ST_IAA tretaments appear to produce the same result regrading survival. Why does no control appear in IAA160 treatment? I stronlgy believe that this Figure can be improved to be more readable.
Intestinal Histopathology, Figure 2: Please indicate lessions caused by the pathogen
Figures 4: control greatly variable and only ST IA160 and ST Con results are grouped. In the same manner great variability of results in Figure 3. Do these variable per group results provide a significant difference in SPSS?
Figure 7: Only 180 treatment favors firmicutes ... please discuss the possible reason. Also The Firmicutes/Bacteroidetes (F/B) ratio is widely accepted to have an important influence in maintaining normal intestinal homeostasis. A decreased F/B ratio is usually observed with inflammatory bowel disease (IBD). Hence estimation of the ratio might provide further evidence for your hypothesis.
Author Response
We feel great thanks for your professional review work on our article. As you are concerned, there are several problems that need to be addressed. According to your nice suggestions, we have made extensive corrections to our previous draft, the detailed corrections are listed below.
Comments 1: Introduction: IAA presents biological functions in many organisms and has been studied mostly in plants since it acts as a plant growth hormone. Please refer in more detail on the production, possible bacterial catabolism and roles of IAA in the host gut and the common gut bacteria.
Response 1: IAA can be produced by a variety of bacteria in the host gut, such as Bacteroides fragilis, Bacteroides thetaiotaomicron, Bacteroides eggerthii, Bacteroides ovatus, Clostridium paraputrificum, Clostridium sticklandii, Clostridium bartlettii, Clostridium putrefaciens, Clostridium lituseburense, Clostridium saccharolyticum, Bifidobacterium adolescentis, Bifidobacterium longum subsp. Longum, Bifidobacterium pseudolongum, Eubacterium hallii, Eubacterium cylindroides, Parabacteroides distasonis, and Peptostreptococcus asscharolyticus. These bacteria can metabolize dietary tryptophan, converting it into indole-3-aldehyde, indole-3-acetic acid, serotonin, indole-3-acetamide, indole-3-acetonitrile, and then converting it to IAA, or directly metabolizing ortho aminobenzoic acid to produce indole-3-acetonitrile, which is then converted to IAA.
Direct studies of IAA in vivo have shown that IAA can mediate the AHR pathway, inhibit inflammation, regulate immunity and intestinal mucosal homeostasis, and improve the therapeutic effect of pancreatic cancer in a non-AHR-dependent manner. In vitro studies have shown that IAA can promote the antioxidant and anti-inflammatory functions of cells, among others. The research of IAA in vivo and in vitro and its mechanisms are becoming more and more clear. We are extremely delighted that our research direction - IAA is attracting more and more attention.
Comments 2: Results: Figure 1: more details are needed in legend. Can B, C, D, E be combined in one graph? Also, all ST_IAA tretaments appear to produce the same result regrading survival. Why does no control appear in IAA160 treatment? I stronlgy believe that this Figure can be improved to be more readable.
Response 2: Thanks for your suggestion, we have summarized Figure 1B-E in one graph and added data details for easy reading. We hypothesized that due to the appropriate concentration of IAA, ST_IAA_40 and ST_IAA_80 had the same effect on improving survival. While 160 mg/kg IAA may have toxic effects on mice, although ST_IAA_160 has a significant regulatory effect on intestinal microbiota, the improvement of intestinal barrier function and inflammation in this group is minimal. We will consider further mechanistic studies.
Comments 3: Intestinal Histopathology, Figure 2: Please indicate lessions caused by the pathogen
Response 3: Thanks for your suggestion, we have added yellow arrows to the pictures of intestinal pathological sections to mark inflammatory cell infiltration and goblet cells decrease caused by the pathogen.
Comments 4: Figures 4: control greatly variable and only ST IA160 and ST Con results are grouped. In the same manner great variability of results in Figure 3. Do these variable per group results provide a significant difference in SPSS?
Response 4: There are significant differences in the analysis of these results in SPSS. Your consideration is reasonable, the difference within the group is slightly larger. We will pay attention to the details of the test operation to improve such problems in future experiments. Thank you for your reminding.
Comments 5: Figure 7: Only 180 treatment favors firmicutes ... please discuss the possible reason. Also The Firmicutes/Bacteroidetes (F/B) ratio is widely accepted to have an important influence in maintaining normal intestinal homeostasis. A decreased F/B ratio is usually observed with inflammatory bowel disease (IBD). Hence estimation of the ratio might provide further evidence for your hypothesis.
Response 5: We used the homogenized microbial data to calculate the relative abundance ratio of Firmicutes to Bacteroidetes. However, the abundance of Bacteroides in the ST challenge group declined too low, and the variation of the ratio within the group was too large to be analyzed and mapped. Although we conducted a preliminary experiment prior to this study, selecting a significant but moderate concentration of ST challenge, the significant inhibition of Bacteroidetes abundance by the pathogen infection was still strong. We will consider changes in the microbial community later when designing the concentration and time of the test.
Thanks to the professional comments again that point out the above problems. We hope these explanations would answer your doubts. And we hope you will find our revised manuscript acceptable for publication.

Reviewer 2 Report
Comments and Suggestions for Authors
I enjoyed reading this paper although I am not a "mouse" person (but I am very interested in the scientific field). However, I find the formatting of Table 1 confusing. I do not know what "gland bubble expansion" is, nor is it apparent to me what the symbols used (slashes, hyphens) signify. Please clarify.
I do have some ethical concerns about the use of mice in this type of experimentation. There is not much new discovery or mechanistic enquiry. This study is consistent with, and integrates, much existing information and in that context this is useful. However, is it worth the suffering and killing of the animals involved? I would encourage the authors to use their expertise in a more humane line of enquiry in future.
Author Response
Thank you for your interest in our manuscript. We have considered the comments carefully and tried our best to revised the manuscript accordingly. Our responses are given in a point-by-point manner below. All the revisions have been addressed in the revised manuscript shown in red.
Comments 1: I find the formatting of Table 1 confusing. I do not know what "gland bubble expansion" is, nor is it apparent to me what the symbols used (slashes, hyphens) signify. Please clarify.
Response 1: Thank you for your reminding. We have amended "gland bubble expansion" to "Acinar dilation" in both Table 1 and Figure 2. "-" hyphens stands for "No pathological lesions", which has been replaced with "No lesions" in the manuscript. Slashes indicates that this indicator is not rated at a higher level, and notes have been made at the bottom of the table.
Comments 2: I do have some ethical concerns about the use of mice in this type of experimentation. There is not much new discovery or mechanistic enquiry. This study is consistent with, and integrates, much existing information and in that context this is useful. However, is it worth the suffering and killing of the animals involved? I would encourage the authors to use their expertise in a more humane line of enquiry in future.
Response 2: On the one hand, the experiment was approved by the Institutional Animal Care and Use Committee of China Agricultural University (Beijing, AW32013202-1-1). All protocols were performed according to the relevant standards of animal welfare of China Agricultural University. On the other hand, our study is the first to show IAA's mitigation of Salmonella infection in vivo, which is the primary food-source pathogen, thus our study has potentially important implications for the improvement of intestinal health in animals and humans. We will do our best to improve animal testing and reduce animal suffering, and we will use our expertise in a more humane line of enquiry in future.
Thank you again for your positive and constructive comments and suggestions on our manuscript. We hope you will find our revised manuscript acceptable for publication.

Reviewer 3 Report
Comments and Suggestions for Authors
The manuscript describes an interesting study aimed at verifying the role of IAA in a treatment against Salmonella infection in an in vivo model. The topic is novel and provides interesting elements to understand the role of tryptophan-derived catabolites generated by the gut microbiota. The study is well structured, the English is generally good and the results need some corrections. On the other hand, some taxonomic and conceptual issues also need to be corrected in the current version, which are detailed below:
- standardize the way in which the bacterial species used are cited: Typhimurium, typhimurium, enterica, Typhi.... On the other hand, "subsp", "serovar", not in italics. Standardize, please
- in the Abstract and text, when the strain is mentioned, its identification in the collection should be added (SL1344). If you write only genus and species, it should be interpreted that all the strains of that genus and d species give the same result, which is an error (strain dependence) - review the entire text: genus, species, families and phyla, in italics as recommended by the American Society for Microbiology (https://journals.asm.org/nomenclature): "Binary names, consisting of a generic name and a specific epithet (e.g., Escherichia coli), should be used for all bacteria. Names of categories at or above the genus level may be used alone, but specific and subspecific epithets may not. A specific epithet must be preceded by a generic name, written out in full the first time it is used in a paper. Thereafter, the generic name should be abbreviated to the initial capital letter (e.g., E. coli), provided there can be no confusion with other genera used in the paper. Names of all bacterial taxa (kingdoms, phyla, classes, orders, families, genera, species, and subspecies) are printed in italics; strain designations and numbers are not.
- Fig. 2: I don't see the title
- Fig. 1: the differences written on line 125-6 are not visible in 1B. Looking at 1B-C-D and E it would seem that the results are identical under all conditions. To clarify
Author Response
Thank you for your affirmation of our manuscript, we will continue to work hard. We have considered the comments carefully and tried our best to revised the manuscript accordingly. Our responses are given in a point-by-point manner below. All the revisions have been addressed in the revised manuscript shown in red.
Comments 1: standardize the way in which the bacterial species used are cited: Typhimurium, typhimurium, enterica, Typhi.... On the other hand, "subsp", "serovar", not in italics. Standardize, please
Response 1: Thank you for your suggestion, we have unified it as “Salmonella typhimurium SL1344” (the following is ST SL1344) in the manuscript. Since ST SL1344 could not be identified in the 16S sequencing results, we retained the current name but modified it format to " Salmonella enterica subspecies enterica serovar Typhi " in the 16S sequencing results in the text and figure.
Comments 2: in the Abstract and text, when the strain is mentioned, its identification in the collection should be added (SL1344). If you write only genus and species, it should be interpreted that all the strains of that genus and d species give the same result, which is an error (strain dependence) - review the entire text: genus, species, families and phyla, in italics as recommended by the American Society for Microbiology (https://journals.asm.org/nomenclature): "Binary names, consisting of a generic name and a specific epithet (e.g., Escherichia coli), should be used for all bacteria. Names of categories at or above the genus level may be used alone, but specific and subspecific epithets may not. A specific epithet must be preceded by a generic name, written out in full the first time it is used in a paper. Thereafter, the generic name should be abbreviated to the initial capital letter (e.g., E. coli), provided there can be no confusion with other genera used in the paper. Names of all bacterial taxa (kingdoms, phyla, classes, orders, families, genera, species, and subspecies) are printed in italics; strain designations and numbers are not.
Response 2: Thank you for your suggestion. Firstly, we have added SL1344 to the abstract and text. In the introduction, the effects of Salmonella are described, as well as the effects of Salmonella typhimurium, not specifically for ST SL1344, so only the strain name has been uniformly modified. Secondly, the format of our paper is formulated according to the double naming method proposed by Swedish naturalist Linnaeus in 1768 and biological naming rules, and according to the requirement of "the scientific name of species above the genus is expressed in Latin, and below the genus (including the genus) is expressed in Latin italics", we have carefully checked and revised the full text, and made changes to the text and figures.
Comments 3: Fig. 2: I don't see the title
Response 3: Fig. 2 is titled: Effects of IAA supplementation on the histopathology of colon and ileum of mice. The text is below the figure 2.
Comments 4: Fig. 1: the differences written on line 125-6 are not visible in 1B. Looking at 1B-C-D and E it would seem that the results are identical under all conditions. To clarify
Response 4: Thanks for your suggestion, we have summarized Figure 1B-E in one graph and added data details for easy reading. We hypothesized that due to the appropriate concentration of IAA, ST_IAA_40 and ST_IAA_80 had the same effect on improving survival. While 160 mg/kg IAA may have toxic effects on mice, although ST_IAA_160 has a significant regulatory effect on intestinal microbiota, the improvement of intestinal barrier function and inflammation in this group is minimal. We will consider further mechanistic studies.
We deeply appreciate your all the valuable comments and suggestions, and look forward to hearing from you regarding our submission. We would be glad to respond to any further questions and comments that you may have. We hope you will find our revised manuscript acceptable for publication.

Round 2
Reviewer 3 Report
Comments and Suggestions for Authors
The authors have responded satisfactorily to all comments and the manuscript appears much improved.